# ID-PreFeR: ID-Preserving Face Restoration with Mixed Data Quality

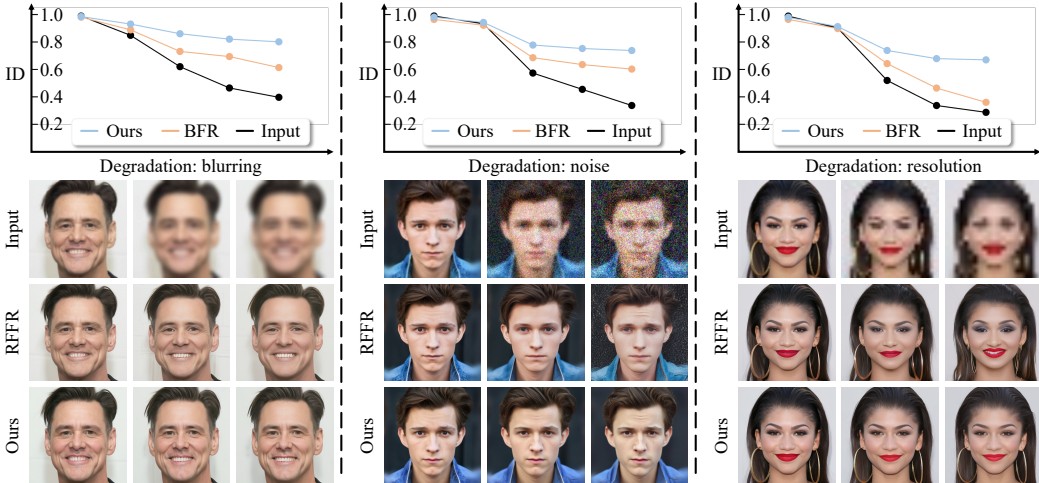

Figure 1: The restoration results of the proposed method and a reference-free face restoration (RFFR) method Liang et al. (2024), with different types of degradation. The more the degradation, the less identity information (measured by ID score Deng et al. (2019)) can be detected from the input image, showing the effectiveness of the proposed method for identity preservation.

## Abstract

This paper introduces ID-PreFeR, a robust ID-preserving face restoration method that addresses the ill-posed face restoration problem by introducing personalized information. Existing methods often suffer from computationally expensive training and storage requirements while being sensitive to the quality of reference images. We present a lightweight personalized injector to enable efficient personalization without the burden of regularization data. Besides, we propose an ID-quality disentanglement training strategy to ensure robust identity learning, even when some of the reference images are of low-quality. An ID-preserving sampling strategy is further proposed to enhance the identity fidelity during inference. Experiments on both synthetic and a newly collected real-world mobile phone dataset validate the effectiveness and practicality of the proposed method.

## 1 Introduction

Low-quality (LQ) facial images, such as small faces in photos taken with mobile phones or unclear portraits found online, are ubiquitous in daily life. Face restoration presents the complex computational problem of reconstructing high-quality (HQ) facial images from those degraded LQ inputs which may be affected by blur, low resolution, noise, and compression artifacts. This task is inherently ill-posed due to multiple plausible solutions for any given LQ input, and is further complicated by the complexity and variability of real-world conditions, including diverse illuminations, poses, and expressions. Given the high sensitivity of the human visual system to facial details, even slight deviations in facial features can become noticeable and potentially alter the perception of identity.

Figure 2: The quality of results from previous personalized face restoration methods deteriorate, if the quality reference images decrease. On the contrary, the proposed method can maintain a robust performance.

As illustrated in Fig. 1, with increasing image degradation levels in the input, the existing blind face restoration method Liang et al. (2024) becomes more likely to drift away from the original identity. Consequently, it is imperative to design a face restoration model that can simultaneously reconstruct realistic facial details while preserving the person's unique identity information.

Previous face restoration methods have significantly improved the visual quality of restored faces, by leveraging generative priors from GANs Ledig et al. (2017); Wang et al. (2018); Shen et al. (2018); Yang et al. (2021), codebooks Esser et al. (2021); Gu et al. (2022); Zhou et al. (2022a), and diffusion models Yang et al. (2024); Wang et al. (2023b); Chen et al. (2024). However, a common drawback stems from the lack of identity information in these pretrained models, resulting in the unfaithfulness to the person's identity and a failure to preserve crucial facial details. To achieve a more faithful restoration for the input identity, early research on personalized face restoration Li et al. (2018; 2020); Dogan et al. (2019); Li et al. (2022) tries to extract instructional facial features from several HQ reference images of the same identity to guide the restoration process. However, as different observations may differ in pose, illumination, make-up, and ornaments, the misalignment can hinder the feature transfer process for previous methods. Recent personalized face restoration methods Varanka et al. (2024); Chari et al. (2023); Ding et al. (2024a) opt to fine-tune diffusion models Rombach et al. (2022) with HQ reference images of the same identity to create personalized neural representations. With the powerful diffusion prior, they are able to achieve better visual quality and identity preservation performance.

There still exist two main challenges for personalized face restoration:

**(1) Heavy personalization burden**. Current personalized face restoration methods Chari et al. (2023); Ding et al. (2024a) often choose to fine-tune the entire parameters of the diffusion models Rombach et al. (2022) to achieve personalization and require the storage of a complete model for each identity. These approaches also necessitate a substantial amount of regularization data that far exceeds the number of reference images to prevent catastrophic forgetting Ruiz et al. (2023) and the consequent loss of model generalization, which poses a significant burden on both training time and storage space.

**(2) Strong HQ dependency**. Existing personalized face restoration methods Li et al. (2022); Nitzan et al. (2022); Varanka et al. (2024); Chari et al. (2023); Ding et al. (2024a) typically learn personalized information from HQ reference images, making their performances highly rely on the quality of reference images. However, in practical applications, it cannot be guaranteed that all acquired reference images will be of high quality. Consequently, as shown in Fig. 2, the image quality of results restored by existing method will be adversely affected if reference images consist of more LQ reference images. Exploring a lightweight and robust personalized face restoration method remains a pressing challenge.

To address the aforementioned challenges, this paper proposes **ID-PrefeR**, a robust **ID-Pre**serving **f**ace **R**estoration method. To relieve the heavy personalization burden, ID-PrefeR learns a

lightweight *personalized injector* by finetuning the weights of cross attention layers with the low-rank adaptation (LoRA) technique Hu et al. (2021a). The personalized injector can be trained on a small number of reference images (typically using 5 images) with a latent prior loss to avoid the requirement on regularization data. Besides, to mitigate the impact of LQ reference images, we introduce an *ID-quality disentanglement* training strategy, which employs a multi-modal large language model Yuan et al. (2021) to incorporate quality-related prompts into the training process, thus enabling the model to focus on learning the person's identity characteristics while ignoring the influence of image quality. We also propose an ID-preserving sampling strategy that utilizes reference images to guide the sampling process, further compacting the hazards of LQ reference images and improving identity preservation. Using only about $1\%$ parameters of the base model Podell et al. (2023), we achieve an efficient ($\sim 4$ minutes on a single L40s GPU) and robust (even in an extreme case that only one reference image is of high quality) personalization process, as shown in Fig. 1 and Fig. 2. Furthermore, considering that the training and evaluation of existing personalized face restoration methods Li et al. (2022); Varanka et al. (2024) often rely on data of celebrities collected online Liu et al. (2015); Li et al. (2022), we collected a dataset using mobile phones to investigate the performance of personalized face restoration methods in real-world scenarios.

The contributions of ID-PreFeR are:

- a personalized injector without regularization data to relieve the heavy optimization burden.

- an ID-quality disentanglement strategy to tackle the dependency on HQ reference images.

- an ID-preserving sampling strategy to further improve identity preservation in test time.

- a real-world dataset captured by mobile phones to facilitate personalized face restoration.

## 2 RELATED WORKS

### 2.1 FACE RESTORATION

As a sub-problem for generic image restoration, face restoration inherently requires the modeling of facial priors for authenticity. Some previous works apply gemetric priors such as segmentation Shen et al. (2020); Chen et al. (2021), facial landmarks Ma et al. (2020); Wang et al. (2022a); Kim et al. (2019); Bulat & Tzimiropoulos (2018), and 3D face shapes Hu et al. (2020; 2021b); Zhu et al. (2022). But these priors are often limited to a high-quality input image, while degradation can make these priors unreliable.

Generative prior is another popular approach. Researchers have turned to GANs Hu et al. (2023); Wang et al. (2021); Chan et al. (2022); Chen et al. (2021); Zhu et al. (2022), vector-quantization codebooks Wang et al. (2022c); Zhou et al. (2022a); Gu et al. (2022), and diffusion models Yue & Loy (2024); Wang et al. (2023b); Yang et al. (2024); Liang et al. (2024) for the distribution of reasonable face images, and develop the restoration process accordingly. Due to the outstanding visual performance, diffusion models have been the focus of recent research. DR2 Wang et al. (2023b) uses a low-pass filter to guide the denoising process of low-quality images. DiffBIR Lin et al. (2024) proposes to first restore and then synthesize details with low-quality image as control Zhang et al. (2023); AuthFace Liang et al. (2024) uses the structure of ControlNet Zhang et al. (2023) and adversarial loss functions to enhance the authenticity in sensitive regions. However, these methods are only conditioned on the LQ input image, and may drift from the input identity.

Given several reference images, personalized face restoration methods restore faces with more individual-specific details. By warping the reference image to the input image, GWAINet Dogan et al. (2019) uses the warped image to guide restoration. ASFFNet Li et al. (2020) requires facial landmarks to select the most similar reference image. DMDNet Li et al. (2022) builds a transformer-based dictionary and improves restoration by aligning the features of the input and reference images. PFStorer Varanka et al. (2024) finetunes personalization blocks when training ControlNet Zhang et al. (2023) on reference images. However, these methods face challenges when reference images contain LQ images. The proposed method learns a neural identity representation using an ID-quality disentanglement strategy, relieving the requirements on HQ reference images.

## 2.2 TEXT-TO-IMAGE PERSONALIZATION

Text-to-image synthesis aims to generate images according to the prompts provided by users, while text-to-image personalization further refines the generated images to match the identity of a specific concept embedded in the reference images. Some recent methods finetune parts of the model on the reference images Ruiz et al. (2023); Kumari et al. (2023b); Wei et al. (2023); Avrahami et al. (2023); Ye et al. (2023), and others only optimize the inverted word embedding for the concept Gal et al. (2022); Avrahami et al. (2023); Chen et al. (2023), and still others add more conditioning Voynov et al. (2023); Zhang et al. (2023) or employ the prompt-to-prompt techniques Hertz et al. (2022); Brooks et al. (2023) to steer the generation during sampling.

However, these methods either require a number of regularization images to prevent language drifting problem Ruiz et al. (2023), or are computationally expensive in practice Gal et al. (2022); Avrahami et al. (2023). We aim to lift the computation burden by only finetuning the LoRA Hu et al. (2021a) of the cross attention modules in the text-to-image model, and address the need for regularization data by using a frozen initial model.

## 3 METHODOLOGY

The task of ID-preserving face restoration requires several portrait images of an individual as reference, and the goal is to restore low-quality portrait images while keeping the identity of the subject. As an overview, the training and inference pipeline of ID-PreFeR is illustrated in Fig. 3.

We first briefly introduce the preliminaries of diffusion models in Sec. 3.1. Then we propose a lightweight *personalized injector* that only modifies a fraction of the parameters related to text conditions in the noise prediction network, detailed in Sec. 3.2. In this way, the personalized injector can cooperate with other pretrained image restoration models and boost identity preservation, achieving plug-and-play personalized face restoration. To alleviate the dependency on high quality (HQ) reference images, we propose an *ID-quality disentanglement* strategy in Sec. 3.3 to exploit the information from mixed-quality images. Finally, to preserve the identity even when few HQ images are given, we propose an ID-preserving sampling in Sec. 3.4, which allows for a more explicit control over the synthesized portrait.

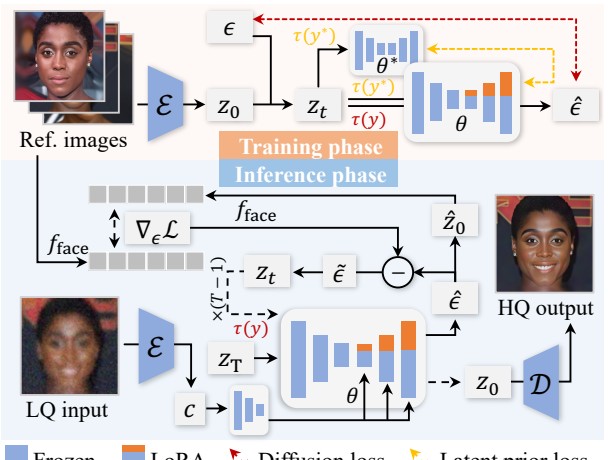

Figure 3: Overview of the training and inference pipeline of ID-PreFeR. We obtain a lightweight personalized injector by training a LoRA for cross attention decoder layers in the noise prediction network $\epsilon_\theta$ with a diffusion loss (using text embedding $\tau(y)$ from prompts with identifiers) and a latent prior loss (using text embedding $\tau(y^*)$ from prompts without identifiers). During inference, the ID-preserving sampling is applied by optimizing the predicted noise in each denoising step, thus the face feature vector estimated from the denoised image correlates with reference images.

## 3.1 PRELIMINARIES

**Diffusion models.** To summarize the diffusion models Wang et al. (2022b); Ho & Zhou (2021) in a few words, they aim to learn a mapping from the noise distribution to a desired image distribution. With a fixed forward process that adds noise to an image, diffusion models learn a denoising process that removes noise to approximate the original image. The denoising process is performed with a U-Net or Transformer parameterized by $\theta$, and the predicted noise is denoted as $\epsilon_\theta$. It is usually conditioned on an intermediate result $x_t$ in the process of denoising, and the current timestep $t$. To control the generated results towards any desired distribution, the input prompts $y$ are turned into

text embeddings $\tau(y)$ with a pretrained text encoder $\tau$, and serve as a condition for the denoising process as well.

For efficiency, diffusion models are adapted to the latent domain Rombach et al. (2022), where an encoder $\mathcal{E}$ first maps the input image $x$ into a latent representation $z = \mathcal{E}(x)$ with a lower resolution but richer information, and a decoder $\mathcal{D}$ maps $z$ back into pixel domain as $x' = \mathcal{D}(z)$. The latent diffusion model (LDM) operates on the latent $z$ and is trained with the following diffusion loss:

$$\mathcal{L}_{\text{LDM}} = \|\epsilon - \epsilon_\theta (z_t, t, \tau(y))\|_2^2, \tag{1}$$

and the diffusion process and denoising process is given by

$$z_t = \sqrt{\bar{\alpha}_t} z + \sqrt{1 - \bar{\alpha}_t} \epsilon, \ \hat{z} = \frac{z_t - \sqrt{1 - \bar{\alpha}_t} \epsilon_\theta(z_t, t, \tau(y))}{\sqrt{\bar{\alpha}_t}}, \tag{2}$$

where $\epsilon \sim \mathcal{N}(0, 1)$ is the added noise to $z_t$, and $z_t$ is the noised latent representation at timestep $t$. $\hat{z}$ is the estimated denoised latent representation, while $\alpha_t$ is the scaling factors that control the noise level at each timestep. $\bar{\alpha}_t = \prod_{i=1}^{t} \alpha_i$ is the cumulative scaling factor.

ControlNet Zhang et al. (2023) extends LDM by adding extra conditioning signals derived from input image processing, such as edge maps or segmentation masks. It freezes the noise prediction network and trains a copy of the encoder part. The trainable encoder is initiated with the original weights, and a zero-convolution is added between its output and the decoder, enabling it to leverage the original Stable Diffusion model's knowledge while guided by another condition $c$. The loss function for ControlNet is:

$$\mathcal{L}_{\text{Ctrl}} = \|\epsilon - \epsilon_\theta (z_t, t, \tau(y), c)\|_2^2. \tag{3}$$

A naive solution for general image restoration is to condition the denoising process of ControlNet on the LQ image, as done by previous methods Liang et al. (2024); Lin et al. (2024). However, they are still severely under-determined, and tend to drift from the actual identity.

### 3.2 PERSONALIZED INJECTOR

The diversity of plausible face distribution calls for the design to inject the identity prior into the denoising process. Inspired by previous works on personalization Ruiz et al. (2023); Kumari et al. (2023a), we finetune cross attention layers of the decoder part in the noise prediction network on the provided reference images with the LoRA Hu et al. (2021a) technique, and optimize with the diffusion loss defined in Eq. (1). However, the finetuning inevitably hinders the generative capability by causing the language drift problem Lee et al. (2019); Ruiz et al. (2023), though the change is limited to a low rank.

To preserve the generative prior, previous works Ruiz et al. (2023) constrain the training with regularization images, by supervising on the images from the class where the subject belongs. More recent works have alleviated the need for regularization image, and turned to the original diffusion model weights $\theta^*$ for supervision. Based on the intuition that using the class prompts without the personalized token should yield a similar response, researchers have tried to supervise on intermediate representations, such as the text embedding Wang et al. (2024), the attention response Zhang et al. (2024), the noise prediction results Wu et al. (2024). Due to the implicit nature of attention response and text embedding, we preserve the generative prior by regularizing on the predicted noise with a latent prior loss:

$$\mathcal{L}_{\text{reg}} = \|\epsilon_\theta(z_t, t, \tau(y^*)) - \epsilon_{\theta^*}(z_t, t, \tau(y^*))\|_2 \tag{4}$$

where $y$ is the prompts with the identifier token, and $y^*$ is the prompts for the class. For example, if $y$ is "a photo of a [V] face", $y^*$ should be "a photo of a face". The regularization is applied in the latent domain for a more fine-grained control, since the distance in the latent domain is more aligned with the pixel error than that in attention response.

### 3.3 ID-QUALITY DISENTANGLEMENT

To learn the identity from several photos precisely, each one of them counts. However, the quality of these reference images can vary significantly, ranging from high-quality (HQ) portraits with clear

details to low-quality (LQ) images degraded by noise, blur, or compression artifacts. If directly optimized on LQ images, the model may learn the artifacts as a property of the identity to personalize. To address this challenge, we propose an ID-quality disentanglement strategy which aims to separate the identity information from the quality degradation present in the reference images.

We try to view the disentanglement from another prospective: The LQ input image can be seen as a degraded version of an HQ image, defined by $z_{\text{LQ}}^{(i)} = z_{\text{HQ}}^{(i)} + \epsilon_{\text{deg}}^{(i)}$, where $\epsilon_{\text{deg}}^{(i)}$ is the unknown degradation noise. Eq. (2) can thus be rewritten into a noise combination as

$$z_t = \sqrt{\bar{\alpha}_t}z + \sqrt{1-\bar{\alpha}_t}(\epsilon + \omega_t\epsilon_{\text{deg}}), \tag{5}$$

where $\omega_t = (\frac{\bar{\alpha}_t}{1-\bar{\alpha}_t})^{\frac{1}{2}}$. Since the diffusion models are trained to fully denoise the image, the output of the pretrained backbone should be close to a slightly altered noise $\epsilon + \omega_t\epsilon_{\text{deg}}$, instead of $\epsilon$.

To fix this discrepancy, and to separating the degradation from the identity information, we propose to add some quality-aware pseudo word as learnable tokens to the prompts $y$. Initialized according to the quality of the reference image, the quality-aware pseudo words $\{Q_i\}$ are jointly optimized with $[V]$, such that $\{Q_i\}$ compensates for $\epsilon_{\text{deg}}$ caused by image degradation. In inference they are removed from the prompts, so that only the identity-related $[V]$ determines the output.

Meanwhile, we employ the Min-SNR-$\gamma$ weighting during training, and Eq. (1) is modified to

$$\mathcal{L}_{\text{LDM}} = \min\left(\gamma \cdot \omega_t^{-2}, 1\right) \cdot \|\epsilon - \epsilon_\theta(z_t, t, \tau(y))\|_2^2, \tag{6}$$

where $\gamma = 5$ is a fixed hyperparameter. Note that the factor $\omega_t$ is the signal-to-noise ratio, which can be neglected on a high timestep, when $\epsilon$ is significantly larger than $\omega_t\epsilon_{\text{deg}}$. Thus the Min-SNR-$\gamma$ weighting ensures that the degradation noise in the image itself can be neglected, when the timestep is low and signal-to-noise ratio is high, and offers a more robust training Hang et al. (2023).

### 3.4 ID-PRESERVING SAMPLING

Despite the ID-quality disentanglement strategy, too few HQ reference images can still hinder the performance, as the degradation in LQ images makes the desired data distribution less specific. To mitigate this, we propose an ID-preserving sampling to guide the denoising process with identity information.

In each denoising step, the predicted noise is optimized for several iterations, such that the predicted image has a similar facial identity vector to a reference HQ image from the training set. The facial identity vector is extracted by a pretrained face recognition model Deng et al. (2019). The optimization process in the ID-preserving sampling can be formulated as

$$\tilde{\epsilon} = \hat{\epsilon} + \delta\nabla_\epsilon \left\langle f_{\text{face}}\left(\mathcal{D}(\frac{z_t - \sqrt{1-\bar{\alpha}_t}\epsilon}{\sqrt{\bar{\alpha}_t}})\right), f_{\text{face}}(x_{\text{ref}}) \right\rangle, \tag{7}$$

where $\delta$ is the step size to update the noise prediction result, and $x_{\text{ref}}$ is a selected HQ reference image in the training set, with respect to the similarity between it and the LQ input. $f_{\text{face}}$ is a pretrained face recognition model used to extract facial features. The cosine similarity $\langle \cdot, \cdot \rangle$ between facial features is used as the loss function to optimize the noise prediction result.

This approach effectively counteracts the potential identity loss during the denoising process, tresulting in more accurate and identity-preserving restorations.

## 4 EXPERIMENTS

### 4.1 EXPERIMENTAL DETAILS

**Baselines.** For personalized face restoration methods, we select a CNN-based model, DMDNet Li et al. (2022), and two of the most recent diffusion-based methods, FaceMe Liu et al. (2025) and Gen2Res Ding et al. (2024b). In addition, we consider GFPGAN Wang et al. (2021), Code-Former Zhou et al. (2022b), DiffBIR Lin et al. (2024), and Authface Liang et al. (2024), which are representative blind face restoration methods, with various generative priors applied.

Table 1: Quantitative comparisons of the proposed method with 4 blind face restoration methods (*i.e.*, GFPGAN Wang et al. (2021), Authface Liang et al. (2024), CodeFormer Zhou et al. (2022b), and DiffBIR Lin et al. (2024)) and the personalized face restoration methods of DMDNet Li et al. (2022), FaceMe Liu et al. (2025), and Gen2Res Ding et al. (2024b). The specifications of reference images (Ref. spec.) are listed. ↑ (↓) denotes higher (lower) values indicate better performance. Numbers colored in red indicate the best performance, while blue for the second best. The metrics of the input and ground truth are for reference.

| Method | Ref. spec. | PSNR↑ | SSIM↑ | LPIPS↓ | CLIPIQA↑ | MANIQA↑ | ID↑ |
|--------|-----------|-------|-------|--------|----------|---------|-----|
| GFPGAN | × | 25.50 | 0.748 | 0.301 | 0.503 | 0.449 | 0.613 |
| CodeFormer | × | 25.07 | 0.726 | 0.388 | 0.598 | 0.429 | 0.586 |
| DiffBIR | × | 25.29 | 0.690 | 0.401 | 0.615 | 0.504 | 0.562 |
| AuthFace | × | 25.32 | 0.683 | 0.294 | 0.699 | 0.643 | 0.649 |
| DMDNet | HQ | 25.04 | 0.712 | 0.316 | 0.613 | 0.491 | 0.635 |
| FaceMe | LQ & HQ | 25.82 | 0.724 | 0.265 | 0.646 | 0.504 | 0.704 |
| Gen2Res | LQ & HQ | 24.30 | 0.748 | 0.424 | 0.491 | 0.331 | 0.446 |
| Ours | LQ & HQ | 25.51 | 0.729 | 0.242 | 0.701 | 0.649 | 0.784 |
| Input | N/A | 24.58 | 0.714 | 0.672 | 0.309 | 0.144 | 0.613 |
| Ground truth | N/A | $+\infty$ | 1.000 | 0.000 | 0.650 | 0.546 | 1.000 |

**Datasets.** The base model of the proposed method is trained on FFHQ Karras et al. (2021) and Unsplash dataset Chesser et al. (2020). Following previous works Li et al. (2022); Varanka et al. (2024), quantitative evaluations are conducted on CelebRef dataset Li et al. (2022). For each identity, we randomly select 5 images as reference data, and uses the rest of the images for evaluation. Following DMDNet Li et al. (2022), we apply various degradations to mimic the real-world scenarios. We swap some of the HQ reference images with their degraded versions, to discuss the impact of the number of HQ reference images (#HQ) in the experiment.

We also include some real-world images captured by mobile phones, to evaluate the in-the-wild performance of the proposed method and baselines. The degraded faces are acquired by capturing in challenging conditions, such as low light, long distance, motion blur, and compression.

**Metrics.** We measure the pixel-wise, structural, and perceptual similarity of the predicted images and the ground truths, by calculating PSNR, SSIM, LPIPS Zhang et al. (2018). For the specific task of face restoration, we follow previous works and compute the identity similarity (ID) Deng et al. (2019) which calculates the cosine distance of face feature vectors. We also list the reference-free image quality metrics of CLIPIQA Wang et al. (2023a) and MANIQA Yang et al. (2022) for reference of the image quality.

### 4.2 QUANTITATIVE RESULTS

Tab. 1 shows the quantitative comparisons of the baseline methods and the proposed method, where we also include the metrics of input LQ images and ground truths. Note that input images already have a high PSNR and SSIM performance, almost surpassing baseline methods, despite the other metrics indicating their poor quality.

The fidelity is further proved by the lowest LPIPS score, which indicates the generated results are similar to ground truths perceptually. The highest ID score shows that ID-PreFeR indeed preserves the identity information better than the baselines. ID-PreFeR also achieves the best CLIPIQA and MANIQA scores, indicating its best quality and photorealism. Combined with the qualitative results, the proposed method outperforms previous state-of-the-art methods in terms of fidelity, identity preservation, and quality.

### 4.3 QUALITATIVE RESULTS

As human vision system is naturally more aware of the face details, we illustrate qualitative results to show the efficacy of the proposed method. For an unbiased and informed comparison, we include cases of various skin colors. Due to the page limit and the inferior performances shown in Tab. 1, the

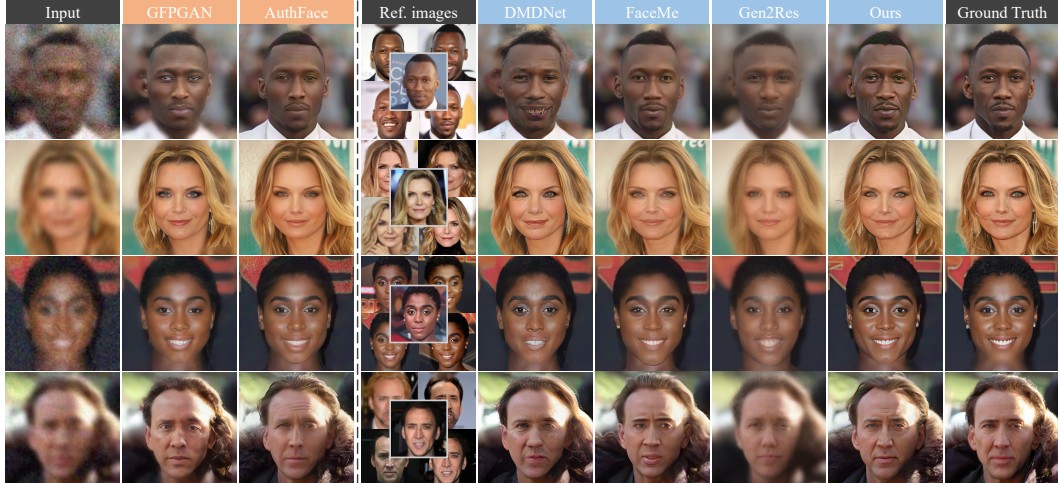

Figure 4: Restoration results on synthetic degraded images, compared with reference-free face restoration baselines (orange titles) and personalized face restoration baselines (blue title, using a combination of HQ and LQ reference images).

Table 2: Quantitative ablation study. Numbers in red indicate the best performance, while blue for the second best. "CA" and "SA" refers to the cross attention and self attention layers, respectively.

| Method | #Params | PSNR↑ | SSIM↑ | LPIPS↓ | CLIPIQA↑ | MANIQA↑ | ID↑ |
|---|---|---|---|---|---|---|---|
| W/o prior preservation | 6.41 M | 25.52 | 0.731 | 0.245 | 0.699 | 0.624 | 0.763 |
| W/o ID-quality disent. | 6.41 M | 25.57 | 0.698 | 0.279 | 0.674 | 0.555 | 0.775 |
| W/o ID-preserving samp. | 6.41 M | 25.07 | 0.696 | 0.281 | 0.666 | 0.554 | 0.750 |
| LoRA all CA | 11.82 M | 25.34 | 0.709 | 0.289 | 0.621 | 0.489 | 0.762 |
| LoRA all CA & SA | 12.57 M | 25.45 | 0.720 | 0.290 | 0.599 | 0.472 | 0.773 |
| Ours | 6.41 M | 25.51 | 0.729 | 0.242 | 0.701 | 0.649 | 0.784 |

results of CodeFormer Zhou et al. (2022b) and DiffBIR Lin et al. (2024) are omitted, as they show worse performance compared to other blind face restoration methods.

In Fig. 4, we show the results on synthetic data regarded from CelebRef Li et al. (2022), and randomly use a combination of HQ and LQ reference images (#HQ=3 in average). The reference-free face restoration methods can generate realistic images, with Authface Liang et al. (2024) slightly better than GFPGAN Wang et al. (2021) in terms of photorealism, but still slightly deviate from the ground truths for identity preservation. For personalized face restoration methods, DMDNet Li et al. (2022) tends to have trouble locating the facial landmarks, such as the mouth and ears in the examples, while Gen2Res Ding et al. (2024b) has trouble capturing the identity. FaceMe Liu et al. (2025) generates slightly altered appearance, such as the emotionless look in the first example and the hair artifacts in the last example, which makes the results weird. In contrast, the proposed method does not need to align the face components, and generates more realistic images with better identity preservation compared with baselines.

In the top 2 rows of Fig. 5, we show the restoration results on images collected from the Internet. The input LQ images have gone through various kinds of compression, so the degradation pattern should differ a lot from the training data. GFPGAN Wang et al. (2021) and AuthFace Liang et al. (2024) are able to generate realistic faces, though somehow different from the reference images in terms of identity. DMDNet Li et al. (2022) and FaceMe Liu et al. (2025) sometimes fail to align the face in the input LQ images, and thus the generated images look broken or fake. Gen2Res Ding et al. (2024b) tends to modify the identity. In comparison, ID-PreFeR outperforms the baselines in terms of identity preservation, as well as realistic synthesis.

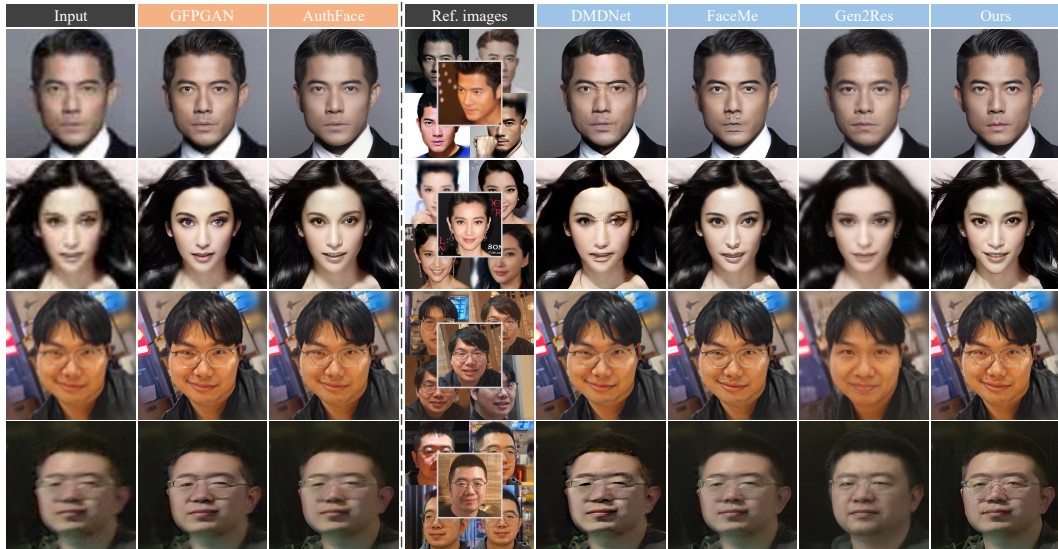

Figure 5: Restoration results of degraded images from the Internet (top 2 rows) and captured by mobile phones (bottom 2 rows).

In the bottom 2 rows of Fig. 5, we show results on real-world degradations. The input images are selected from video frames captured by mobile phones, in low-light moving scenes. The input images thus present mixed degradations, including noise, blur, low resolution, and compressio. GFP-GAN Wang et al. (2021) and DMDNet Li et al. (2022) tend to keep the original pixel-wise correspondence, even when the compression distorts the input image. AuthFace Liang et al. (2024) and Gen2Res Ding et al. (2024b) generates more realistic images, but the identity-related features are less preserved, such as the beard and the shape of the eyes in the two examples. FaceMe Liu et al. (2025) manages to learn the identity by aligning the facial landmarks, but fails when the input suffers from severe compression or distortion, as observed in the shown examples. In comparison, the proposed method manages to restore the identity and the subtle facial expression faithfully in both examples. The eyes as well as the shapes of the glasses are preserved, even when the glasses are absent in some reference images. Due to page limit, please refer to Sec. B for more results.

## 4.4 Ablation Study

To validate the effectiveness of the proposed method, we conduct an ablation study on different components. The quantitative results of the ablation study is shown in Tab. 2. Note that the complete model only has 6.41M parameters, demonstrating its lightweight property compared with the setting of optimizing the LoRA of all cross attention layers (CA) or all attention layers (CA & SA). Its superior performance on the majority of metrics (except PSNR and SSIM) also demonstrates the effectiveness of the proposed modules. Though better PSNR and SSIM performances do not necessarily indicate better visual quality Zhang et al. (2018), the highest ID score still verifies the efficacy of the proposed design. This is further validated by visual comparisons, detailed in Sec. A, where the complete ID-PreFeR generates images with higher fidelity and fewer artifacts.

## 5 Conclusion

In this work, we propose ID-PreFeR, a novel framework that personalizes the portrait restoration process, given a few reference images with mixed data quality. It can disentangle the contents from varying image quality through joint optimization, enabling the model to only focus on the shared identity information in finetuning. Data and source codes will be open source upon acceptance.

**Limitations.** For reference images, the variations in makeup, jewelry, and glasses may lead to feature cancellation during training, since the model aims to learn the shared identity information. ID-PreFeR may show diverse results in these features, rather than adhering to any single reference.

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

# A    QUALITATIVE ABLATION STUDIES

We try lifting the latent prior loss in Eq. (4) and present the results in Fig. 6. We notice that the white balance leans towards magenta, and more artifacts of the wrinkles can be observed with fewer reference images. The color drift from the input shows that applying the latent prior loss can anchor the denoising process to faithfully restore the input LQ images, instead of over-fitting to the biased distribution of the reference images.

We also try removing the quality cues and the min-SNR-$\gamma$ weighting in the ID-quality disentanglement strategy, without which the results look blurry when using few HQ images. As shown in Fig. 7, the wrinkles and facial details look much blurrier when using only 1 HQ reference image, compared to the complete model.

Then we ablate the ID-preserving sampling strategy, and the results are shown in Fig. 8. The vanilla sampling discards the iris color which is a key identity cue, while the ID sampling preserves the brown pupil, with a similar color to the ground truth.

# B    MORE VISUAL COMPARISONS

In addition, we provide some more qualitative comparisons, as presented in Fig. 9 and Fig. 10. Fig. 9 compares the performances of the baselines and ID-PreFeR on synthetic scenes, where the proposed method achieves the most close restoration to the ground truth. Fig. 10 shows web photos of some celebrities which have undergone unknown mixed compression. The proposed method is able to restore their well-known facial identity information.

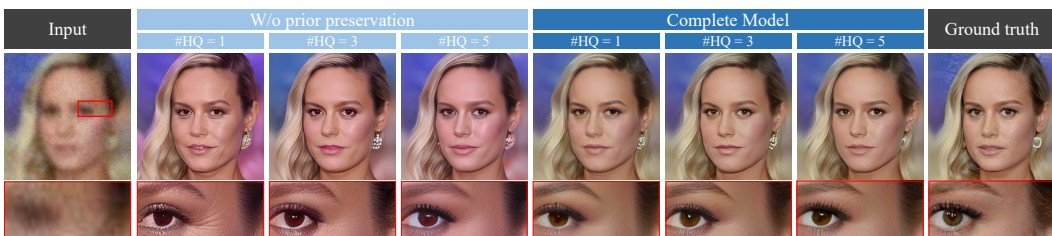

Figure 6: Ablation study on the latent prior loss. Without prior preservation loss, the tone changes and details of the faces vary in this example.

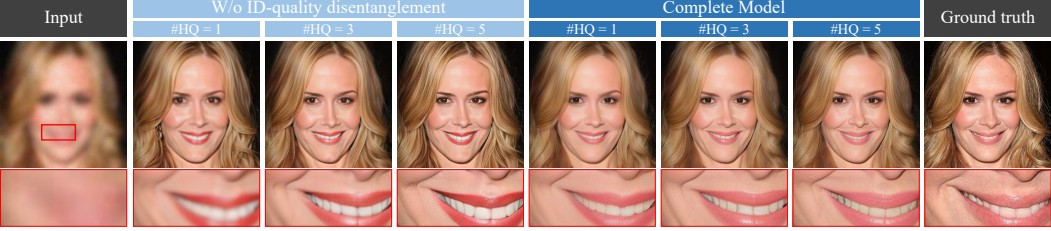

Figure 7: Ablation study on the ID-quality disentanglement strategy. Without ID-quality disentanglement, the image quality deteriorates and some details are lost.

# C    ETHICAL IMPACT

## C.1    DEMOGRAPHIC FAIRNESS

Face restoration systems are highly sensitive to visual identity cues that vary with ethnicity, age, and other demographic factors. Without careful consideration, such systems may show uneven performance across population subgroups. Note that ID-PreFeR is designed to focus on identity-preserving features while minimizing dependence on lighting and skin-tone–related signals, as validated in the

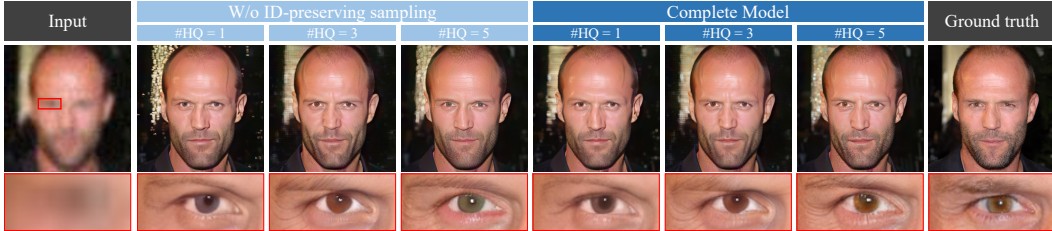

Figure 8: Ablation study on the ID-preserving sampling. With ID-preserving sampling, the model is able to generate the faces with the desired identity more deterministically, as observed from the eye color.

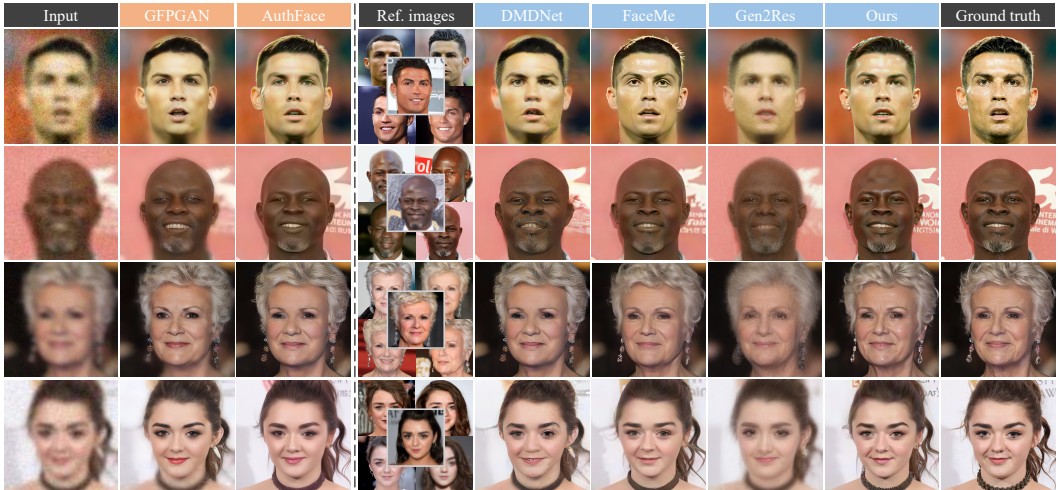

Figure 9: More qualitative comparisons on ID-preserving face restoration. The input images contain synthetic degradation, and the ground truths are listed in the final column.

diverse qualitative cases throughout the paper. It is able to maintain consistent restoration quality across ethnic groups. However, small residual disparities remain, and it shows the need for continued research in this area.

## C.2 PRIVACY AND MISUSE CONCERNS

With the marvel of technical advances often comes more concerns about misuse. The capacity to reconstruct high-fidelity facial images from degraded inputs introduces potential misuse risks. In particular, applications involving surveillance, de-anonymity, or forensic prediction without consent are explicitly discouraged. We will urge users to follow legal frameworks and ethical standards, especially in contexts involving vulnerable or marginalized populations.

## C.3 TRANSPARENCY AND ACCOUNTABILITY

We support responsible deployment by releasing model evaluation protocols and encouraging third-party audits on fairness and generalization. Further, we advocate for the development of community benchmarks that better capture demographic variability and downstream societal effects.

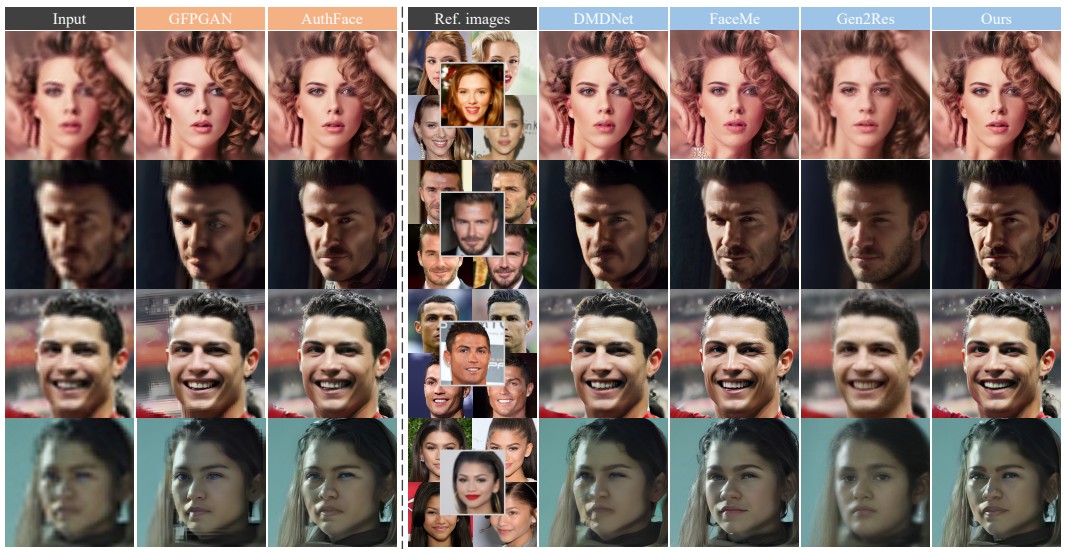

Figure 10: More qualitative comparisons on ID-preserving face restoration. The input images are collect from Internet, with compression and various degradations. Ground truths are not available in this setting.

