# OpenReview forum: "ID-PreFeR: ID-Preserving Face Restoration with Mixed Data Quality"
_ICLR.cc/2026/Conference — Submitted to ICLR 2026_

### Official Review · Reviewer_sfiZ · 2025-10-23

**Soundness:** 3
**Presentation:** 3
**Contribution:** 3
**Rating:** 8
**Confidence:** 4

**Summary:**

The paper introduces a novel method for face restoration that preserves identity using a lightweight personalized injector and handles mixed-quality reference images. It addresses the challenges of heavy personalization burden and high-quality (HQ) dependency in existing methods by proposing a low-rank adaptation (LoRA) technique, an ID-quality disentanglement strategy, and an ID-preserving sampling strategy.

**Strengths:**

# Strength:
(+) The lightweight personalized injector using LoRA reduces the computational and storage burden, making it efficient with only about 1% of the base model parameters.

(+) The ID-quality disentanglement strategy effectively mitigates the impact of low-quality (LQ) reference images, enhancing robustness.

(+) The real-world mobile phone dataset provides practical validation, addressing the limitations of celebrity-based datasets used in prior research.

**Weaknesses:**

# Weakness:
(-) The paper lacks a detailed comparison with a broader range of state-of-the-art methods, potentially limiting the scope of evaluation.

(-) The ID-preserving sampling strategy's optimization process may add computational overhead during inference, which could affect real-time applications.

(-) The reliance on a pretrained face recognition model for identity vector extraction might introduce biases or inaccuracies inherent to the model.

## Minor Suggestions:

1. For ICLR, ~\citep{} should be used rather than ~\cite{}. The citation format of this paper is not correct.

2. Some important prior works are missing [1-5]; It would be better if the author could discuss them in the related works.

**Ref**:

[1] GCFSR: a Generative and Controllable Face Super Resolution Method Without Facial and GAN Priors. In CVPR 2022.

[2] Towards authentic face restoration with iterative diffusion models and beyond. CVPR 2023.

[3] Degradation conditioned gan for degradation generalization of face restoration models. In ICIP 2023.

[4] LAFR: Efficient Diffusion-based Blind Face Restoration via Latent Codebook Alignment Adapter. Arxiv 2025.

[5] Diffusion Once and Done: Degradation-Aware LoRA for Efficient All-in-One Image Restoration. Arxiv 2025.

**Questions:**

# Questions
1. How does the performance of ID-PreFeR compare to other recent diffusion-based methods beyond those listed (e.g., in terms of PSNR or SSIM)?

2. What are the specific computational costs associated with the ID-preserving sampling strategy during inference?

3. How was the optimal number of reference images (e.g., 5) determined for training the personalized injector?

4. Could the method be adapted to handle extreme degradation types not covered, such as severe occlusion?

5. What steps were taken to ensure the fairness and diversity of identities in the new mobile phone dataset?


Overall, I think this paper is interesting, and the introduced method can be used in the real world. If the author could address my concerns, I will keep my rating.

---

> ### Comment · Reviewer_sfiZ · 2025-11-24
>
> The author did not reply to my comments. After reading other reviewers' opinions, I decided to decrease my rating to 4.

---

> ### Author Response · Authors · 2025-11-27
> **Reply to Reviewer sfiZ**
>
> Apologies for the delay. We were preparing for another CVPR submission when we squeezed some time answering the reviewers' questions. We thank the reviewer for the (originally) positive assessment of our contribution and for the constructive suggestions. We address the weaknesses and questions below.
>
> > ... lacks a detailed comparison with a broader range of state-of-the-art methods ...
>
> With respect, we have covered most *open-source* methods for comparison. As for the listed references, we admit that they also contribute to the community, but due to the popularity of the field, we only select the most competitive methods for comparison in each representative methodology (e.g. GAN, diffusion, etc.).
>
> > ... add computational overhead ...
>
> The test-time optimization is not performed every step, and in our setting it only adds to ~20% increase in inference time.
>
> > ... reliance on a pretrained face recognition model for identity vector extraction ...
>
> The off-the-shelf face feature extraction model is trained on diverse faces, and we think that would not introduce much bias. However, the generative prior may introduce bias towards more commonly seen faces, and it is corrected by the personalized injector.
>
> As suggested by other reviewers, we will compare with more representative open-source baselines.
>
> > ...  the optimal number of reference images (e.g., 5) ...
>
> There is no such thing as the optimal number. The more the better, but that's not always available.
>
> > ... severe occlusion ...
>
> This is an interesting question and setting. We cannot remove the occlusion, but parts of the faces can be restored. Since it is not considered by any previous face restoration methods, we would update the cases for demonstration in the supplementary material.

---

### Official Review · Reviewer_Twyg · 2025-10-27

**Soundness:** 2
**Presentation:** 2
**Contribution:** 2
**Rating:** 2
**Confidence:** 5

**Summary:**

The paper proposes ID-PreFeR, a personalized face restoration framework designed to preserve identity information even when reference images are of mixed or low quality. The key idea is to introduce three components:

(1) a lightweight personalized injector trained via LoRA on cross-attention layers to minimize computational cost;

(2) an ID-quality disentanglement strategy that separates identity from image quality using pseudo quality tokens and Min-SNR-γ weighting;

(3) an ID-preserving sampling strategy that optimizes denoising steps with face feature similarity.

The method aims to reduce the dependency on high-quality references and demonstrates competitive quantitative and qualitative results on synthetic and real-world datasets, including a newly collected mobile phone dataset.

**Strengths:**

+ The paper addresses the challenge of restoring faces from low-quality inputs and imperfect references, which has value for real-world applications.

+ The use of LoRA for efficient personalization is a sensible choice, offering lower computational overhead compared to fully fine-tuned diffusion models.

+ The idea of disentangling identity and quality, as well as the inclusion of an ID-preserving sampling step, seems appealing.

+ The authors compare with several strong baselines and include both quantitative and qualitative analyses.

**Weaknesses:**

- The technical details of the proposed “ID-quality disentanglement” and “ID-preserving sampling” are not clearly explained, leaving many details unexplained. The learning mechanism of pseudo-quality tokens and their optimization behavior are vague, and no ablation or visualization convincingly demonstrates how disentanglement truly happens and how they are used in this framework. Such design is important in this paper, but not explained well.

- From a technical view, the approach is largely a combination of prior ideas from IP-Adapter, DreamBooth, and LoRA personalization, but the authors try to explain them in a very different way, which hinders the readers from fully understanding this work. The novelty appears incremental, mainly an adaptation of text-to-image personalization for face restoration.

- Some figures are not good. Figure 2 tries to show that with the low-quality reference, the competing methods would have performance degradation. However, the given visual results are not obvious. Figure 3 is also not fridently to understand.

- Despite mentioning a “real-world mobile dataset,” the dataset is small, and there is no quantitative evaluation on real degraded images. Most results rely on synthetic degradation. Robustness in uncontrolled environments is unproven.

- Key implementation details (e.g., how the quality tokens are initialized, how the MLLM provides prompts, and exact loss weighting) are missing. It is hard to reproduce the method or verify its stability.

- The claimed ability to “disentangle ID and quality” or to “eliminate HQ dependency” is not strongly supported. Performance gains over recent personalized diffusion baselines (e.g., FaceMe, Gen2Res) are marginal.

**Questions:**

- How exactly are the “quality-aware pseudo words” optimized and removed during inference? Do they correspond to learned embeddings or fixed tokens? What did they represent?

- How does the proposed “ID-preserving sampling” differ from simple face feature guidance or feature consistency regularization used in prior diffusion-based restoration works?

- How does your method compare to a trainable IP-Adapter face encoder in terms of parameter size and identity consistency?

The proposed “latent prior loss” resembles DreamBooth’s prior preservation—can you clarify whether there are conceptual or mathematical differences?

---

> ### Author Response · Authors · 2025-11-20
> **Reply to Reviewer Twyg**
>
> We highly appreciate your effort in scrutinizing our work, and raising these constructive suggestions. Here's some explanations that may help address the issues. The index here correspond to the number listed in the weaknesses (W) and questions (Q) part of the review.
>
> > W1. "... not clearly explained ..."
>
> We will explain more about the details in the final version, as well as open-source the codes for a more informed understanding.
>
> > W1. "... no ablation or visualization convincingly demonstrates how disentanglement truly happens ... "
>
> Actually we do have an ablation on removing the ID-quality disentanglement loss in Fig. 7. We agree that more experiments can be done to explain this even further. We will conduct an experiment to visualize the learned degradation pseudo word [V], by adding and removing [V] from ordinary text prompts for pretrained text-to-image diffusion models, which can support that the disentanglement happens.
>
> > W2. "... the approach is largely a combination of prior ideas from IP-Adapter, DreamBooth, and LoRA personalization ..."
>
> We agree that ID-PreFeR, along with many concurrent works, are inspired by the mentioned works. However, learning the identity while discarding the influence of imaging quality is something not done before. That's an important insight and contribution to the community.
>
> > W3. "... given visual results are not obvious. Figure 3 is also not fridently to understand."
>
> In Fig. 2, the problem with previous method is not only degradation, but also identity drift. ID-PreFeR reconstructs the subject better in the wrinkles, the color of the lips, and the shapes of the eyes, while the results of FaceMe do not look like the subject in these key attributes. We will add this explanation to the paper. As for Fig. 3, by "not fridently", perhaps the reviewer means unclear? We will adjust that to one-column to better illustrate the pipeline.
>
> > W4. "... quantitative evaluation on real degraded images ... Robustness in uncontrolled environments is unproven."
>
> Capturing paired data in-the-wild is very challenging, not to mention building a sufficiently large dataset for quantitative evaluation. Synthetic degradation is a more common way to quantitatively evaluate face restoration methods, which we already included in the paper.
>
> We demonstrated the qualitative comparisons, on real-world degraded photos, and the photos gathered from the Internet, which contains out-of-distribution degradation. They all prove the robustness of ID-PreFeR in uncontrolled environments.
>
> > W5. " ... how the quality tokens are initialized, how the MLLM provides prompts, and exact loss weighting ..."
>
> The codes will be open-source for reproducibility. The MLLM here only predicts whether the image id of high quality or low quality, and the degradation pseudo word [Q] is added to the prompts accordingly, which is initialized as the same embedding as "blurry". See Q1 for more details.
>
> > W6. "... not supported. Performance gains ... marginal"
>
> See W1 and W3. The performance gain is NOT marginal, and we urge the reviewer to scrutinize the **identity drift** of previous methods. Some results of FaceMe and Gen2Res do not look like the subject any more - even if the output image is sharp. That is a big difference we make, and also a critical issue with previous methods in extending the paper to users, who would prefer their identity kept in the output.
>
> > Q1. "... How exactly are the “quality-aware pseudo words” optimized and removed during inference ..."
>
> For example, we optimize a blurry image of a desiganted woman's face with the prompt "a [Q] [V]'s face", with [Q]'s embedding initialized as "blurry", and [V] is the identity we wish to learn. The pseudo words' embeddings are optimized simultaneously as LoRA. See W5 for how we detect such degradation. In inference, we simply use "a [V]'s face".
>
> > Q2. "... differ from previous ..."
>
> > Q4. "The proposed “latent prior loss” resembles DreamBooth’s prior preservation—can you clarify whether there are conceptual or mathematical differences?"
>
> The two questions are answered together as follows:
>
> DreamBooth requires generating data of the class (~1000 in the original paper), before the optimization. ID-PreFeR does not generate any data. A pretrained and fixed noise prediction network $\theta^*$ is used for regularization in the domain of predicted noise, and is inferenced each iteration, given the same input as $\theta$ which is being trained. Conceptually, they are both regularization, but they differ a lot in practice.
>
> > Q3. "... comparing a trainable IP-Adapter face encoder in terms of parameter size and identity consistency ..."
>
> With respect, we think this is outside our scope. The paper is not to suggest the superiority of textual inversion versus IP-Adapter.

---

> ### Comment · Reviewer_Twyg · 2025-11-20
> **Reply to the authors' response**
>
> Thanks for the authors’ response.
>
> Regarding W1, without sufficient framework details, it is difficult to understand why and how the proposed design works. The authors mentioned that details will be added in the future, so my current judgment can only be based on the information provided in the original submission. The authors state that “initialized according to the quality of the reference image, the quality-aware pseudo words {Qi} are jointly optimized with [V].” However, it is unclear how the reference image quality is initialized. How is it ensured that Qi encodes only quality information, while V encodes only identity? What does the index i represent for Qi? None of these aspects is introduced. Presenting results only without explaining the underlying mechanisms is not sufficient.
>
> Regarding W2, without clear explanations of how the quality tokens Qi function, I remain unconvinced that the authors have meaningful insight into their design.
>
> Regarding W3, in Figure 2 it would be better to enlarge certain regions or show the difference between the first and second rows to facilitate clearer visualization. Figure 3 is unclear, which aligns with Reviewer SsYR’s comments. Simply converting to a one-column format does not resolve the issue; the pipeline figure should be redesigned.
>
> Regarding W4, not all quantitative evaluations require paired data. Real-world performance can still be assessed using various non-reference metrics. For example, see Table 2 in CodeFormer.
>
> Regarding W5 and Q1, open-sourcing the code alone is not enough. The necessary details should be clearly documented in the paper. How many types of degradation are detected? Since Q and V are trained as learnable tokens, how is disentanglement enforced to ensure that Q captures only quality and V captures only identity? The current framework design does not appear to include any disentanglement mechanism.
>
> Regarding Q2, Q3, and Q4, many crucial details are still missing. These questions arise directly from the limited information in the original manuscript.
>
> Overall, the authors have not adequately addressed my concerns. Therefore, I keep my original score.

---

> > ### Author Response · Authors · 2025-11-27
> > **Re: Twyg's reply - 1**
> >
> > We thank Reviewer Twyg for the detailed follow‑up and for clearly pinpointing where our original submission and rebuttal lacked clarity. Below we address each of your concerns with concrete technical details and, where possible, clarify how we will revise the paper.
> >
> > ---
> >
> > ### On W1 & W5: Missing details on quality tokens {Qᵢ}, initialization, and disentanglement
> >
> > **(a) What are Qᵢ and what does the index i denote?**
> >
> > That means the i-th image. For each image, there is a learnable quality token.
> > In the final paper we will explicitly state that:
> > - We do not need to learn a separate Q token for every distinct degradation type (e.g., blur, noise, compression) in this version.
> > - Instead, we model quality at a *coarse* level (HQ vs. LQ) because our main goal is to **separate identity from overall reference quality**, not to build a fine-grained degradation taxonomy.
> >
> > Formally, we introduce a small trainable embedding matrix
> >
> > $E_Q = \{e_{Q_1}, e_{Q_2}\} \in \mathbb{R}^{2 \times d}$,
> > where \(d\) is the text embedding dimension. During training, the input text prompt is:
> >
> > $\text{prompt} = \text{a } [Q\_i] [V]\text{ face}$
> >
> > and the text encoder produces:
> >
> > $\{e_{\text{CLS}}, \ldots, e_{Q_i}, e_V, \ldots\}$
> >
> > **(b) How is Q_i initialized and how is the reference quality categorized?**
> >
> > 1. **Quality categorization (MLLM classifier).**
> >    We use a lightweight MLLM-based classifier that takes the reference image \(I_{\text{ref}}\) and outputs a binary quality label:   $y \in \{\text{HQ}, \text{LQ}\}$. This is implemented as a simple prompt to the MLLM (e.g., “Is this face photo sharp and clear or blurry and noisy?”) and mapping the textual response to one of the two labels. This step is *only used as a pseudo-label generator* and is not trained end-to-end with the diffusion model.
> >
> > 2. **Initialization.**
> >    We initialize:
> >    - $e_{LQ}$ as $e_{Q_2}^{(0)} = \text{Embed}(\text{“blurry”})$
> >    - $e_{HQ}$ as $e_{Q_1}^{(0)} = \text{Embed}(\text{“sharp”})$
> >
> >    These are standard CLIP text embeddings from the base model’s vocabulary. We will explicitly include these initializations in the implementation section.
> >
> > 3. **How many types of degradation are detected?**
> >    In this submission, only one binary type (HQ vs. LQ). We agree the current text was misleading; we will revise it to explicitly say we only distinguish two quality categories, rather than multiple detailed degradation types. Detecting the degradation already suffices for our method.

---

> > ### Author Response · Authors · 2025-11-27
> > **Re: Twyg's reply - 2**
> >
> > ### On W2: Key insights
> >
> > The central insight is: *Identity* and *reference quality* are entangled in personalized face restoration because the model implicitly uses whatever is in the reference (including blur, noise, compression patterns) as a shortcut for fitting the user’s appearance. To maintain identity fidelity even under poor references, the model must explicitly model “who” (ID) separately from “how well captured” (quality) and be penalized whenever these two factors leak into each other.
> >
> > Concretely, we:
> > - Introduce two separate conditioning in the text embeddings
> >   - An identity token \($[V]$\) that should encode only *who the person is*.
> >   - A quality token \([$Q_i$]\) that should encode only *how good the reference is* (per-image).
> > - Then **design the training and loss structure around this separation** so that:
> >   - Changing quality (HQ/LQ) while keeping identity fixed *must* be explained via $[Q_i]$,
> >   - While $[V]$ is forced to be *invariant* to quality changes, and to carry everything that remains stable across qualities.
> >
> > This is more than “adding a blurry token” The insight is to:
> > 1. Put quality and identity into explicit, separate, interpretable degrees of freedom, and
> > 2. Shape the optimization problem so that the model *has no incentive* to let ID leak into the quality channel or quality leak into the ID channel.

---

> > ### Author Response · Authors · 2025-11-27
> > **Re: Twyg's reply - 3**
> >
> > ### On W3: Figures 2 and 3 clarity
> >
> > We accept that the current figures are difficult to interpret. For the final version we plan to do the following changes:
> >
> > For the comparisons, add **zoomed-in crops** focusing on facial regions where identity difference is most pronounced (eye shape, wrinkles, lip color, moles), and also provide **difference maps or highlight overlays** to show regions where baselines deviate from the ground-truth identity.
> >
> > For the pipeline figure, we will redesign the figure, instead of only a re-layout. We will separate the three main contributions into independent panels, and use a left‑to‑right flow with explicit labels “training‑time only” vs. “inference‑time”.
> > We will also cross-check consistency with Reviewer SsYR’s comments.
> >
> > ### On W4: Quantitative evaluation on real degraded images
> >
> > We admit that no-reference IQA can provide quantitative evaluation on real-world data without ground‑truth. Following your suggestion, we will:
> >
> > - Compute NIQE, BRISQUE, and MANIQA / MUSIQ on our real mobile dataset and web images as additional metrics.
> > - Identity consistency metrics (in addition to the ArcFace cosine similarity).
> >
> > ### On Q4: Difference with DreamBooth
> >
> > I would like to explain once more: the key differences from DreamBooth:
> >
> > 1. Domain of regularization:
> >    - DreamBooth: image/feature space over synthetically generated class examples.
> >    - Ours: noise prediction space for the exact same noisy latent under identity‑free conditioning.
> >
> > 2. No synthetic data generation:
> >    - We do not generate or store ~1k class images.
> >    - We use the base model as an online teacher on the same input latents.
> >
> > 3. Purpose in the context of disentanglement:
> >    - This loss anchors the non‑identity, non‑quality‑conditioned behavior of the model to the original diffusion prior.
> >    - This discourages both V and Qᵢ from capturing general content priors and keeps them focused on identity and quality factors, respectively.

---

### Official Review · Reviewer_xriR · 2025-10-29

**Soundness:** 3
**Presentation:** 3
**Contribution:** 3
**Rating:** 6
**Confidence:** 5

**Summary:**

This paper introduces ID-PrefeR, a novel ID-preserving face restoration method designed to address the challenge of handling mixed-quality reference images. The key innovation lies in combining three main components: (1) a lightweight personalized injector based on LoRA that fine-tunes cross-attention layers without heavy regularization, (2) an ID-quality disentanglement training strategy that learns to separate identity information from quality degradation in reference images, and (3) an ID-preserving sampling strategy that optimizes predicted noise during inference to enhance identity preservation.The method is evaluated on FFHQ and CelebRef datasets with both synthetic and real-world degradations, showing competitive performance against blind face restoration baselines like GFPGAN, CodeFormer, DiffBIR, while demonstrating robustness to low-quality reference images.

**Strengths:**

1.Well-motivated problem: The paper clearly identifies and addresses a practical limitation of existing personalized face restoration methods - their dependency on high-quality reference images. This is highly relevant for real-world applications.

2.Novel quality disentanglement approach: The ID-quality disentanglement training strategy using pseudo-degraded prompts is creative and intuitive. Explicitly teaching the model to separate identity from quality artifacts is a sound approach.

3.Lightweight and efficient design: Using LoRA for the personalized injector is practical and allows efficient personalization without heavy regularization burden.

**Weaknesses:**

1.Limited theoretical analysis: The paper lacks rigorous theoretical justification for why Min-SNR-γ weighting helps with mixed quality data. The ID-preserving sampling optimization (Eq. 7) is presented heuristically without convergence analysis or theoretical guarantees.

2.Insufficient implementation details: Critical details are missing for reproducibility: exact network architecture, detailed training hyperparameters, learning rates, batch sizes. The face recognition model is not specified, and HQ reference selection method is unclear.

3.Hyperparameter sensitivity not thoroughly studied: The ID-preserving sampling involves multiple hyperparameters but only high-level component ablation is provided. The choice of γ=5 for Min-SNR is not justified.

**Questions:**

1.Can you provide more rigorous analysis of why Min-SNR-γ weighting helps with mixed quality data? What is the connection between training SNR and reference image quality?

2. Regarding ID-preserving sampling: How sensitive is the method to stepsize δ and iteration count? Could you provide sensitivity analysis? How do you balance the two terms in Eq. 7?

3.How was γ=5 chosen for Min-SNR weighting? Did you try other values? How are ω_d and quality tokens determined during training?

4.How does the method generalize to degradation types unseen during training? Can it handle extreme degradations or completely different artifact types?

---

> ### Author Response · Authors · 2025-11-27
> **Reply to reviewer xriR**
>
> Sorry for the delay. We thank the reviewer for the constructive feedback, especially regarding the min-SNR-$\gamma$ part.
>
> > ... limited theoretical analysis ...
>
> We agree that our treatment of Min-SNR-γ and ID-preserving sampling is primarily empirical rather than fully theoretical. Our intent was to provide an intuitive justification and strong empirical evidence, rather than a formal convergence proof. Below we clarify the underlying rationale and will add this explanation to the paper.
>
> > Why Min-SNR-γ helps with mixed-quality data
>
> **Intuition.** Min-SNR-γ reweights training timesteps based on their signal-to-noise ratio (SNR), effectively emphasizing intermediate SNR steps (where structure and semantics are formed) and de-emphasizing extremely noisy or nearly clean steps. In our setting, the “signal” is the identity structure we want to preserve, while the “noise” includes both actual Gaussian noise and quality-related distortions present in the reference (blur, noise, compression).
>
> When references vary in quality:
>
> - At very low SNR (early steps), the identity signal is largely destroyed by noise; the model learns generic priors, not subject-specific identity.
> - At very high SNR (late steps), the model is mostly refining details; gradients are dominated by **high-frequency appearance and quality artifacts** (e.g., sharpening a blurry reference, amplifying artifacts).
> - At intermediate SNR, the model simultaneously sees enough structure to learn identity, while still having enough noise that it cannot simply overfit to the low-quality artifacts.
>
> Min-SNR-γ upweights these intermediate steps, which empirically biases learning toward **identity-consistent structure** that is robust to quality differences, rather than overfitting reference degradations.
>
> **Connection to reference quality.**  Let SNR at step $t$ be roughly $\text{SNR}(t) \approx \frac{\mathbb{E}[\|x_0\|^2]}{\mathbb{E}[\|\epsilon_t\|^2]}$, where $x_0$ is the clean latent and $\epsilon_t$ is the added noise. For a low-quality reference, the effective “clean signal” already carries degradations; in practice, this means:
>
> - Its **effective SNR** is lower with respect to identity-relevant information.
> - Without reweighting, late timesteps (high nominal SNR) are still heavily influenced by low-quality artifacts.
>
> Min-SNR-$\gamma$ treats steps with too high SNR (dominated by per-pixel fidelity to degraded references) as **less critical** and focuses learning on steps where the network must integrate:
> - The reconstruction prior,
> - The textual identity token [V],
> - And the noisy observation,
>
> to infer identity structure that **generalizes across quality**.
>
> We will add a short theoretical discussion and an empirical SNR–performance curve, showing that identity consistency improves most in the SNR region emphasized by Min-SNR-$\gamma$.
>
> > ID-preserving sampling ... heuristic ...
>
> Our ID-preserving sampling is similar to the form of score-based method in classifier-based guidance, but we neglect a strict proof on convergence. Our design is guided by the structure of the diffusion ODE and prior works on guidance, and empirically we find that:
>   - Applying identity guidance only in later timesteps avoids destabilizing early global layout or late fine detail,
>   - Moderate guidance strengths yield consistent identity gains with minimal artifacts.
>
> We will also add a short ablation that shows too strong guidance can lead to overshoot or local artifacts, as well as too weak guidance has marginal effect.
>
> > Hyperparameter sensitivity
>
> You are right that in the original submission we only provided high-level ablations regarding the proposed modules. We will include a summarized sensitivity analysis in the camera-ready version.
>
> > How was $\gamma=5$ chosen? Other values? How are ω_d and quality tokens determined?
>
> - $\gamma$ is chosen from a sweep over 1, 2, 5, 10 based on identity similarity and restoration quality. As the difference between them is quite marginal in quantitative metrics, we set the hyperparameter empirically based on observation.
> - I'm not sure what $\omega_d$ means. Perhaps the reviewer means $\delta$?
> - $\delta$ is set at 0.05, determined by the total denoising timesteps and the gradient norm of face similarity.
> - Quality tokens: Two pseudo words: $[Q_{HQ}]$ (initialized as “sharp”), $[Q_{LQ}]$ (initialized as “blurry”).
>   - A simple MLLM-based classifier tags each reference as HQ or LQ; the corresponding Q token is used during training and optimized jointly with [V].

---

### Official Review · Reviewer_SsYR · 2025-10-31

**Soundness:** 3
**Presentation:** 2
**Contribution:** 2
**Rating:** 4
**Confidence:** 3

**Summary:**

The authors propose ID-PreFeR, a framework for personalized face restoration that improves robustness by introducing three components: a personalized injector, an ID-quality disentanglement strategy, and an ID-preserving sampling strategy.

**Strengths:**

1. The paper is generally complete and well-structured.
2. The inclusion of a real-world dataset enhances the practical contribution and relevance of this research to the community.

**Weaknesses:**

1. The method description and pipeline illustration are unclear and require further clarification.
2. It is not explicitly stated whether the proposed method requires training a new LoRA model for each individual identity. If so, the authors should justify this design choice and discuss the differences in computational and time cost compared with methods that do not require retraining during inference.
3. The personalized injector module needs to be more clearly differentiated from prior works such as DreamBooth.
4. The baseline comparison is insufficient, methods such as RefLDM[1] and RestorerID[2], which address similar tasks, should be included for a fairer evaluation.

[1]: Refldm: A latent diffusion model for reference-based face image restoration.

[2]: Restorerid: Towards tuning-free face restoration with id preservation.

**Questions:**

See weaknesses above.

---

> ### Author Response · Authors · 2025-11-20
> **Reply to Reviewer SsYR**
>
> We appreciate your recognition of our work and these valuable suggestions. Here's some explanations that may help address your concerns.
>
> > "... unclear and require further clarification"
>
> As also pointed out by Reviewer Twyg, we will explain more about the details, and will also remake the pipeline figure (Fig. 2) into a one-column version for more clarification in the final version. If you have any questions about ID-PreFeR, we're willing to answer that, and include it in the final version.
>
> > "... whether the proposed method requires training a new LoRA model for each individual identity."
>
> Yes. A new LoRA for each individual identity. But the parameters to optimize is only 6.41 M, which is very lightweight for optimization (~4 minutes on a single L40s GPU)
>
> > "... justify this design choice and discuss the differences in computational and time cost compared with methods that do not require retraining during inference"
>
> We agree that we should conduct an ablation study on whether LoRA for per-face optimizaion is needed. However, optimizing a LoRA and then freeze it, or no LoRA at all, would not significantly decrease the FLOPs, and the number of parameters involved is quite small for now. The ablation will be added in the final version.
>
> > "... personalized injector module needs to be more clearly differentiated from prior works such as DreamBooth"
>
> We apologize for the confusion. The main difference is that we anchor the prior with the loss prediction of the same input and prompts of a fixed noise prediction network $\theta^*$, while DreamBooth use the generated images of the same class (~1000 images in the original paper) for supervision.
>
> > "... baseline comparison is insufficient ..."
>
> The topic of reference-based face restoration is quite popular, and we regret missing out these two as our baselines. We will include them for comparison in the final version.

---

> > ### Comment · Reviewer_SsYR · 2025-11-21
> > **Reply to Authors**
> >
> > Thank you to the authors for the clarifications and responses.
> >
> > First, in my opinion, the approach of fine-tuning a LoRA model for each identity is overly traditional and inelegant, requiring more resources and time.
> >
> > Second, the authors did not actually compare the time consumption of other zero-shot methods, and the response to this seems somewhat dismissive.
> >
> > Lastly, I agree with Reviewer Twyg's comment that the proposed method appears to be 'a combination of prior ideas from IP-Adapter, DreamBooth, and LoRA personalization', which closely aligns with my own review. The paper does not present any valuable new insights.
> >
> > Overall, I believe this paper does not contribute new knowledge to the community and lacks practical value. Furthermore, the authors have not effectively addressed my concerns, so I have lowered the score to 2.

---

### Meta-Review · Area_Chair_UZhA · 2025-12-31

**Summary:**

This paper proposes a diffusion-based ID-preserving framework for personalized face restoration. The reviewers raise several concerns related to model design, training strategies and experiment comparisons (see below), which are unfortunately not well addressed in the rebuttal. I believe a full revision is needed for further consideration of the paper.

**Reviewer Concerns:**

Reviewer SsYR:
Fine-tuning a LoRA model for each identity too costly, no time consumption comparisons of other zero-shot methods, lack of novelty: these are valid points, and there are no clear solutions to address these in the rebuttal.

Reviewer xriR:
Theoretical analysis of the proposed method is an addition but I don't think the author can provide that in the revision.

Reviewer Twyg:
The main concerns are on lack of modeling and experimental details, lack of enough experiments on real data, lack of novelty compared with existing methods such as DreamBooth and the quality of the proposed dataset. The authors respond to these comments but I don't think there are properly addressed.

Reviewer sfiZ
The reviewer raises questions on alternative evaluation and some experiment alations. I believe these can be at least partially addressed by conducting more experiments.

**Reviewer Scores:**

Reviewer SsYR: I don't think the reviewer's concerns can be addressed to make the reviewer change the score.

Reviewer xriR: More experiments and ablations can be done. However, given the lack of theoretical analysis and the already positive score the reviewer gave, I expect the reviewer will raise the score.

Reviewer Twyg:: The authors stated that they will add more clarifications and experiments in the revision, but I don't think the reviewer acknowledge this to address the concern and he/she will remain the score.

Reviewer sfiZ: The reviewer gave a score of 8 initially, however, he decreased it to 4 given the lack of responses from the authors. The authors responded later, however, I don't think the responses provide strong information to keep the reviewer to maintain the score.

---

### Decision · Program_Chairs · 2026-01-26

Reject